# The Clinical and Neuropathological Features of Sporadic (Late-Onset) and Genetic Forms of Alzheimer’s Disease

**DOI:** 10.3390/jcm10194582

**Published:** 2021-10-03

**Authors:** Tanzil Rujeedawa, Eva Carrillo Félez, Isabel C. H. Clare, Juan Fortea, Andre Strydom, Anne-Sophie Rebillat, Antonia Coppus, Johannes Levin, Shahid H. Zaman

**Affiliations:** 1Cambridge Intellectual & Developmental Disabilities Research Group, Department of Psychiatry, University of Cambridge, Cambridge CB2 8PQ, UK; mtr38@cam.ac.uk (T.R.); ecarrife7@gmail.com (E.C.F.); ichc2@medschl.cam.ac.uk (I.C.H.C.); 2Cambridgeshire and Peterborough Foundation NHS Trust, Fulbourn CB21 5EF, UK; 3Sant Pau Memory Unit, Department of Neurology, Hospital de la Santa Creu i Sant Pau, Biomedical Research Institute Sant Pau, Universitat Autònoma de Barcelona, 08193 Barcelona, Spain; JFortea@santpau.cat; 4Center of Biomedical Investigation Network for Neurodegenerative Diseases (CIBERNED), 28031 Madrid, Spain; 5Barcelona Down Medical Center, Fundació Catalana Síndrome de Down, 08029 Barcelona, Spain; 6Department of Forensic and Neurodevelopmental Sciences, Institute of Psychiatry, Psychology and Neuroscience, King’s College London, London SE5 8AF, UK; andre.strydom@kcl.ac.uk; 7South London and the Maudsley NHS Foundation Trust, The LonDowns Consortium, London SE5 8AZ, UK; 8Geriatrics, Institut Jérôme Lejeune, 75015 Paris, France; annesophie.rebillat@institutlejeune.org; 9Department for Primary and Community Care, Department of Primary and Community Care (149 ELG), Radboud University Nijmegen Medical Center, P.O. Box 9101, 6525 GA Nijmegen, The Netherlands; tonnie.Coppus@radboudumc.nl; 10Department of Neurology, Ludwig-Maximilians-Universität München, 80539 Munich, Germany; johannes.levin@med.uni-muenchen; 11German Center for Neurodegenerative Diseases, Feodor-Lynen-Strasse 17, 81377 Munich, Germany; 12Munich Cluster for Systems Neurology (SyNergy), Feodor-Lynen-Strasse 17, 81377 Munich, Germany

**Keywords:** late-onset Alzheimer’s disease, down syndrome, autosomal dominant Alzheimer ’s disease, clinical features, neuropathology

## Abstract

The purpose of this review is to compare and highlight the clinical and pathological aspects of genetic versus acquired Alzheimer’s disease: Down syndrome-associated Alzheimer’s disease in (DSAD) and Autosomal Dominant Alzheimer’s disease (ADAD) are compared with the late-onset form of the disease (LOAD). DSAD and ADAD present in a younger population and are more likely to manifest with non-amnestic (such as dysexecutive function features) in the prodromal phase or neurological features (such as seizures and paralysis) especially in ADAD. The very large variety of mutations associated with ADAD explains the wider range of phenotypes. In the LOAD, age-associated comorbidities explain many of the phenotypic differences.

## 1. Introduction and Background

This review aims to highlight the similarities and differences between the clinical and neuropathological manifestations of Alzheimer’s disease (AD) in sporadic (or late-onset AD) and autosomal dominant AD (ADAD) and Down syndrome-associated AD (DSAD).

AD is a neurodegenerative disease that results in neuronal cell death, causing brain atrophy. The neurodegeneration is thought to be due to the immediate or downstream consequences of the abnormal accumulation of beta-amyloid (Aβ) and hyperphosphorylated Tau proteins that manifest as Aβ neuritic plaques and neurofibrillary tangles (NFTs), respectively. Several other pathological processes also play a role, including the innate immune system, the inflammatory response and mitochondrial dysfunction and the consequent oxidative damage. It is associated with reactive astroglia [1], immune-responsive microglia [2] and neurovascular issues [3]. AD is the most common form of dementia and is characterized by memory loss and cognitive decline in several modalities. In some cases, it can also present with atypical symptoms. AD affects 30–35 million people worldwide and as life expectancy increases, the prediction is that by 2030, it will be experienced by 82 million people at any one time [4].

Late-onset AD (LOAD) is also referred to as sporadic AD and clinically presents in those aged over 65 years. However, there are different subtypes of LOAD and there may be heterogeneities in both the clinical features and neuropathology they demonstrate. AD presenting before 65 years of age is referred to as early onset AD (EOAD). Between 5–7% of all AD presents as EOAD [5]. Though estimates of EOAD usually do not include people with DS, EOAD should comprise both Down syndrome associated AD (DSAD) and autosomal dominant AD (ADAD). However, not all EOAD are DSAD or ADAD.

Amongst all live births, Down syndrome (DS) is the most common aneuploidy. In c.95% of cases the cause is a full trisomy of chromosome 21-the rest are due to partial trisomies, translocations or mosaicisms. DS is associated with growth delays, characteristic facial features, intellectual disability and multiple comorbidities such as congenital heart defects, thyroid dysfunction, autism spectrum disorder, sleep apnoea, hearing loss and visual impairment [6]. All these features, together with the “accelerated ageing” phenotype, used to result in a short life span: in the 1930s, most people with DS lived only up to 10 years of age [7]. This improved lifespan gradually increased to 35 years in the 1980s [8], and nowadays people with DS live on average of 60 years [9]. The increase in longevity has allowed the manifestation and study of AD in DS [10,11]. Virtually all adults with DS over 40 years of age present the typical AD-like neuropathological features of fibrillary (senile) plaques and neurofibrillary tangles (NFTs) in their brains [12]. The prevalence of AD in DS increases with ageing in a more pronounced way, and at a much earlier age than in the general population, or among other groups of people with intellectual disabilities [13]. AD in people with DS is often diagnosed first in young adults and increases exponentially until the majority have a clinical diagnosis of AD around the age of 60 [14,15,16]. According to a twenty year longitudinal study following people with DS [16], 97.4% developed dementia with the risk of dementia increasing from 23% at around 50 to 80% at age 65 and above. However, some people with DS who may have died younger because of other co-morbidities will influence these estimates and so result in a survival bias of the figures.

Autosomal dominant AD (ADAD) is caused by fully penetrant mutations in one of the, thus far recognised, three genes, amyloid precursor protein (APP) and presenilin (PSEN) 1 and 2, which follow a Mendelian autosomal dominant inheritance pattern. ADAD represents < 1% of all AD cases and the relative frequencies due to mutations in PSEN1, PSEN2 and APP are 69%, 2% and 13%, respectively [17]. The number of disease gene mutations discovered is huge: over sixty, seventy and three hundred specific gene mutations are associated with APP, PSEN1 and PSEN2, respectively; knowing the specific mutation for a given case can help inform precision medicine approaches for patients.

APP is encoded on chromosome 21 and is pathological with respect to dementia when it is duplicated(dupAPP) [18]. The dupAPP results in an increase in APP gene dosage, similar to that of trisomy of chromosome 21 in DS.

APP is a transmembrane protein that is proteolytically cleaved at different specific residues by secretases: the α-secretase, which is responsible for the products of the “non-amyloidogenic” pathway, and the β- and γ-secretases that result in the products of the “amyloidogenic” pathway. Cleavage via the β- and γ-secretases at different sites leads to the formation of Aβ peptides of different sizes such as Aβ40 and Aβ42, which are 40 and 42 residues long, respectively. AD causing mutations in the APP gene are clustered around the three cleavage sites, with most of them affecting the γ-secretase site of cleavage. The positions of these mutation sites increases the generation of the Aβ42 which is more insoluble and more prevalent in cored Aβ-plaques compared to the Aβ40 peptide [19]. PSEN is one of the catalytic subunits of the γ-secretase complex [20]. Most PSEN mutations cause a loss of function of γ-secretase and increase the Aβ42/Aβ40 ratio [20], thereby promoting oligomer formation. Aβ peptides undergo a biophysical transformation from monomers to oligomers before being deposited in Aβ plaques. Consistent with the amyloid cascade hypothesis [21], it is such monomers and oligomers that are thought to be particularly toxic to brain tissue. Hereditary forms of AD provides strong support to the amyloid cascade hypothesis as they show how alterations in the processing of APP with resulting aberrant levels of Aβ42 represent a strong driver of synaptic and neuronal loss [20]. However, it is noted that the amyloid cascade hypothesis hinges on neurodevelopmental processes and is therefore best suited to genetic or chromosomal causes of dementia.

When considering ADAD, the phenotype variants between carriers of the different genetic mutations, as well as between variants of the same gene need to be considered [20,22]. ADAD phenotypes are influenced by mutation position and causative gene [22]. The different phenotypes are characterised not only by different clinical features and ages of onset but also by aspects of underlying pathology. For example, APP Flemish and Dutch mutations present with recurrent cerebrovascular events, due to amyloid accumulation in the blood vessels rather than as parenchymal amyloid plaques [23]. In contrast, the APP Icelandic mutation has a protective effect against AD and cognitive decline [24]. With PSEN1, mutations before and after codon 200 are pathologically different [25] and have different ages of onset [22]. PSEN2 mutation carriers show atypical presentations that resemble dementia with Lewy bodies or frontotemporal dementia [26], when compared to the other types of ADAD. The heterogeneity associated with different ADAD mutations is therefore very important to consider.

The different forms of AD have many similarities and differences, understanding of which can allow a deeper insight into the mechanisms of AD. Notably, it can clarify the role that different genes, their mutations and proteins play in the development of the disease. In addition, given the difficulty in the general population of predicting the transition from preclinical to clinical AD, studies of EOAD are valuable as they are expected to increase predictability, are very valuable. Clinical studies in DS have some advantages as there is both a high risk and predictability of developing DSAD, and the AD neuropathology seems more homogeneous than in LOAD. Moreover, in ADAD and DS, the earlier age of onset reduces the impact of confounding factors associated with ageing, thereby allowing the pathological characteristics of AD to be better discriminated. However, the triplicated genes on chromosome 21 may limit the extrapolation of data from DS populations to other populations, and neurodevelopmental factors need to be taken into account.

There are many factors associated with the development of AD. In the non-DS population, these factors include not only ageing, but also cardiovascular risk factors (17), traumatic brain injury [27], number of years of education [28] and genetic risk factors [29]. Genetic risk factors include a family history of AD, including the ADAD mutations in PSEN and APP and the possession of some SNPs or gene alleles that, through the genetic analysis of large populations [30], have been linked with the disease. The possession of the E4 allele of the apolipoprotein E (APOE) gene is one of the major factors that influence the development of AD [31]. The evidence for APOE’s effect on the clinical presentation of ADAD [22,32] and DSAD [33,34,35] is becoming clearer despite genetic mutations that cause AD masking the effect of APOE. For instance, it has been found that the APOE ε4 allele correlated with earlier clinical and biomarker changes of AD in DS [36]. It is also noted that the Christchurch mutation, R136S in APOE3 (homozygous) in a PSEN1-E280A mutation carrier, reported relatively little decline in cognition despite ageing and little evidence of tau-deposition despite relatively greater amounts of amyloid being detected using PET imaging [37].

In DS, an extra copy of APP is sufficient to cause AD [38]. However, there are other genes on chromosome 21 [39] whose overexpression may enhance or modify the risk for AD [40]. Some of these candidate pathogenesis gene modifiers include beta-secretase-2 (BACE2) [27] and DYRK1A [41,42], both of which are related to the calcineurin-NFAT signaling pathway that is altered in AD [43]. Some studies however argue against the activity of BACE2 in DS [44] as being pathological since it is able to cleave APP toward the non-amyloidogenic pathway. Other studies have argued that BACE1 could instead play a greater role [45]. It is also possible that DS cells respond to the increased gene dosage by enhancing DNA methylation [46] thereby accelerating epigenetic changes that are usually associated with ageing [47]. This could account for the “accelerated ageing” phenotype observed in DS.

Comparisons of dementia in LOAD and ADAD [19,48] and in LOAD and DSAD [49] have been carried out. However, there is very limited literature comparing DSAD with ADAD or all three conditions [50]. The aim of this review is to synthesise and present the literature on all the three conditions.

## 2. Methods

A search through PubMed and the Cochrane library was performed to exclude the existence of another review in this topic. Then, a structured review was undertaken following the guidelines provided by PRISMA Protocols [51]. First, a search in PubMed was carried out of the literature published about humans since 2000 and relating to Alzheimer’s disease in Down syndrome, using the MeSH terms: (“Alzheimer Disease” [Mesh] AND “Down Syndrome” [Mesh]). From the 482 results, 219 were assessed for eligibility based on their relevance from their title and abstract, and from those, 133 were selected after a full-text review. For the autosomal form, broader search methods were applied. As there is no MeSH term for this condition, three searches were undertaken with the same parameters as for Down syndrome: (“Alzheimer Disease” [Mesh] AND (clinical features [All Fields])), this produced 340 results from which 27 were screened and 18 selected; (“Neuropathology” [Mesh] AND “Alzheimer Disease” [Mesh]), with 19 results from which 2 were eligible but excluded as duplicated; and (“Histology” [Mesh] AND “Alzheimer Disease” [Mesh]), with 161 results from which 2 were screened and 1 included. We included 36 articles from the 84 publications highlighted by the DIAN study in thie website [52] as their collection of important publications was considered a valuable resource of information for ADAD. DIAN stands for the Dominantly Inherited Alzheimer Network, a longitudinal observational study to monitor individuals who carry one of the gene mutations known to cause ADAD with the main goal of identifying changes and establish reliable biomarkers. For LOAD, no special search was undertaken initially, as the information about this condition is implicit in the articles from the two other conditions. However, some specific searches were carried out to cover omissions in the information, adding a total of 6 articles.

The exclusion criteria applied to select the articles from their abstracts were the following: (a) n = 1, (b) only the title or abstract available, (c) opinion articles, clinical trials, novel test descriptions or protocol revisions, (d) no specific content about AD in papers relating to Down syndrome (e) no specific reference to familial or genetic causes for dementia in the search for ADAD. From the selected articles, we have finally included those that did not fulfil any of the previous exclusion criteria nor, following a full-text review contain any new information. Finally, reference lists of selected papers were also searched for potentially relevant studies, adding 42 papers to the 75 included for Down syndrome, 19 to the 30 included for the autosomal dominant form, and 21 papers to the 7 for the sporadic form of the disease-references from these papers also were examined (an addition of 94). The search process is summarized in Figure 1.

## 3. Clinical Features

Several issues need to be considered for the critical review of the discussion presented below. Comparing inherited and sporadic AD brings different problems, as survival and clinical features are likely to be affected by the age of onset. Younger, healthier patients may survive longer and until the later stages of the disease. Therefore, clinical features may be more noticeable and easily detectable than in older patients with LOAD, who are generally frailer, often with comorbidities that may or may not be related to AD, potentially causing difficulty in diagnosis. Phenotypic heterogeneity, which is especially common in ADAD, also needs to be considered. These differences may be biased by diagnostic issues. In people with DS, there are difficulties arising from the presence of a large range of DS-related comorbidities. There is also great baseline variability in cognitive and adaptive (day-to-day living skills) functioning due to how the intellectual disability interferes with the ability to reliably measure or recognise cognitive and functional decline and which makes the application of standard clinical or neuropsychological tests for a dementia diagnosis harder. In such cases, diagnosis, often based on the caregiver’s report, may be subjective or have interpretative bias such as when more relevance is placed on some signs rather than on others. Finally, first symptoms may pass unrecognized in LOAD while in the familial cases, there can be an earlier awareness to the presentation of symptoms [20]. These factors are critical to consider when comparing the clinical features of the different forms of AD.

### 3.1. Diagnosis

In the general population, there is no single test that allows us to accurately and specifically predict the time of diagnosis of AD before the cognitive and functional decline begins [53]. In fact, a focus of the research in AD is the search for biomarkers that may predict the onset and impact of the disease [54]. However, in ADAD, the full penetrance of the mutations allows the onset of AD to be more predictable. In DS, prevalence data can give an estimate or probability of diagnosis at given age ranges.

### 3.2. Age of Onset

Multiple risk factors influence the age of onset in DS [55], such as certain SNPs (single nucleotide polymorphisms) [56] and genetic variants [57] like APOE, PICALM [58] and even TAU haplotypes [59,60]. Among this group, age of diagnosis is an important predictor of survival time along with severity of intellectual disability, socio-economic status, anti-dementia medication status and history of epilepsy [61].

Among those with DSAD and LOAD, there are sex differences in age of onset, with women at greater risk of developing the condition [62]. Longitudinal studies have suggested that variants in the oestrogen receptor genes-β [63] and -α [64]. Subsequently, it has been suggested that age at menopause increases the risk of AD in DSAD and in LOAD, possibly menopausal oestrogen deficiency being a factor. The impact of oestrogen may be important in familial AD too, since the risk of the condition is associated with in oestrogens receptor-1 gene variants [65].

Figure 2 shows the age of onset of diagnosis and survival time for the different forms of AD. Despite differences in age of onset, individuals with ADAD and LOAD have similar survival times. Among those with DSAD, the time between diagnosis and death in DS may reflect a bias from late diagnosis, or the impact of comorbidities. Such an explanation is consistent with the clinical course of DSAD appearing similar to that of individuals with an APP duplication (dupAPP).

Similarly, shorter survival times [32] are associated with ADAD with clinical symptoms appearing at an earlier or later age, rather than in midlife (between 35 and 65 years) The duration of survival for younger people is likely to reflect the highly pathogenic nature of the mutations [32], while for their older counterparts, it is likely to be limited by expected lifespan. PSEN2 mutation carriers typically have a later onset than the other hereditary cases as shown by a study of 1307 ADAD mutation carriers [32]. This may lead them to be categorised as LOAD [66]. It is important to note that Figure 2 does not account for the differences due to the different mutations that can occur a particular gene in ADAD. For instance, PSEN1 mutations before codon 200 are usually associated with an earlier onset [41.3 ± 7.2] than those with mutations at sites beyond [45.8 ± 6.4], as seen in a retrospective analysis of 168 PSEN1 mutation carriers (*p* < 0.0001) [22].

### 3.3. Vasculature

The vasculature plays a very important role in the pathology of dementia. In LOAD, the association with vascular risk factors may be detrimental as it may accelerate dementia progression by reducing the clearance of Aβ40 across the endothelium [69]. In contrast, in DS, vascular dementia, as well as other vascular pathologies such as infarcts, are uncommon (but see below), perhaps due to an absence of cardiovascular risk factors, such as hypertension [70]. Hypercholesterolemia does not appear to be a risk factor for AD in DS, according to a study of the total serum cholesterol levels of 179 persons with DS [71]. Nevertheless, prescribed statins (to reduce cholesterol) may be protective [72], possibly because they have an anti-inflammatory action. Furthermore, in DS, the integrity of brain blood vessels impacts on dementia pathology in DS [73]. Drachman et al. [73] for example, found that microvascular numbers and endothelial integrity were similar in LOAD and DS [73], thus not going against the microvasculature hypothesis of dementia. Importantly, sleep apnoea and obesity, which are associated with DS, may increase the risk of cerebrovascular pathology [70]. Moreover, some cerebrovascular disease biomarkers such as, enlarged perivascular spaces and white matter hyperintensities are associated with DS in adults [62]. These markers tend to increase as AD progresses, thereby leading us to believe that AD may be associated with cerebrovascular disease. Similar observations are seen in ADAD [74].

Cerebral Amyloid Angiopathy (CAA) is a condition where there are amyloid deposits in the walls of brain vasculature. This can lead to cerebrovascular dysfunction, including blood–brain barrier disruptions and microhaemorrhages, which have been associated with cognitive decline and dementia [75]. CAA is common in LOAD, but it is more prevalent and severe in hereditary cases of AD despite presenting at a younger age, as described in a neuroimaging study [75,76]. The frequency of CAA in ADAD is similar to that in DSAD [76], though this may differ depending on the mutations being compared (see below for comparison with dupAPP). In ADAD, CAA is especially frequent in carriers of mutations in PSEN1 after codon 200 [77]. It is also common in the Flemish APP mutation, which, according to the study of 5 mutation carriers [78], is characterized by cerebrovascular events probably reflecting the fact that in this familial form, senile plaques accumulate mainly on blood vessels, as was seen by image and mass spectrometric Aβ peptide analyses of one family [79]. In DS, CAA is particularly common, especially in middle aged persons [80]. DS provides a good model to study the relationship between AD and vascular problems as certain vascular pathologies are normally rare in DS [81]. The higher frequency of CAA in DS was observed in a study that compared post mortem brain tissue of 32 DS with 80 LOAD [75] and was later confirmed in another study [82]. The study of 5 families showed that CAA is prevalent in people with duplications of the APP locus [83] who present with frequent haemorrhagic events; even more so than in people with DS or APP mutation. The CAA subtype was also different between dupAPP, mutated APP and DS, with more extensive involvement in dupAPP carriers [84]. Conversely, plaque formation was higher in carriers of mutated APP and DS compared to dupAPP [84]. The difference between dupAPP and DS suggests potential protective factors in DS [76,81,83].

### 3.4. Amnestic Phenotypes

AD is characterized by memory problems from the earliest stage of its clinical course. However, this initial feature is not uniform and heterogeneity has been reported to be dependent on age of onset, according to a study of 7815 patients from various academic centres [85].

In DS, the first clinical symptoms appear to be largely similar to those in AD [86] when diagnostic criteria based on those for LOAD in the general population are used. Memory features may be less obvious when diagnosis is wholly dependent on carer reports (see below). In people with DS, early symptoms of AD may be mistaken as part of intellectual disability [87] (‘diagnostic overshadowing’) or may be obscured by comorbidities such as sleep apnoea or depression [88,89,90]. It has been suggested [91], that individuals with higher level of cognitive functioning at baseline are at lower risk of AD but more studies, with larger samples are required.

Starting with a cross-sectional study of 68 institutionalised people with DS, it has been found that one of the earliest signs of dementia is an impairment of recent memory in the absence of loss of long-term memory [92]. The earliest stages of DSAD are characterised by, episodic memory loss [93,94,95,96,97], similar to that of the mild cognitive impairment (MCI) phase described in people with LOAD [98] and ADAD [20]. The loss of episodic memory precedes functional deterioration [99,100] by many years. Similarly, a retrospective study of 449 PSEN E280A mutation carriers described an amnestic cognitive impairment phase preceding the onset of dementia by up to two decades [101]. The loss of episodic memory, measured by free recall correlates with pre-clinical accelerated hippocampal atrophy (see the DIAN cohort studies; [102], and has been associated with early navigation problems [103]. In later stages, a deterioration in cued recall is found [99].

The timeline of deterioration of skills depends on the particular form of AD. In a 10 year longitudinal prospective study of 19 people with ADAD, it was observed that in the early stages, naming, spelling and visuo-perceptual skills were better preserved than in LOAD [104]. Visual deficits occurred in the late stages [20], particularly in PSEN1 mutations after codon 200 [68]. Similarly, for those with DS, loss of visuospatial skills occurs early [93,94,95,96,97]. However, visuo-perceptual deficits, similar to those reported in LOAD [105], have also been reported in people with DS without dementia using psychophysical tests [106] and visual evoked potentials [107]. The mean ages of these DS cohorts were c. 40 years and 26 years, respectively. How these pre-dementia diagnossis changes in DS relate to age-related decline is unclear.

In DS, attention deficits have been described in the early stages of AD [87,97,108], while in ADAD, attention deficits are observed in the preclinical phases of the disease [109]. With regard to language, in DS, there is already a major neurodevelopmental effect on expressive language, and cases of LOAD can also present with language deficits [110]. In fact, assessment of language skills could potentially help in identifying earlier DSAD [111]. In ADAD, language deterioration has been associated with a specific PSEN2 [112] and another PSEN1 mutation, whereas memory has been relatively well-preserved [113].

With AD progression in DS, amnestic symptoms increase [114]. Carers report general slowness, greater effects on cognitive domains, and the loss of adaptive skills [49,115] over time. In contrast, among DS individuals without dementia, a decline in functional skills is much more common [116], perhaps reflecting better established skills in individuals who do not have dementia and therefore not so demanding.

### 3.5. Non-Amnestic Phenotypes

In AD, non-amnestic phenotypes are rarer than amnestic phenotypes. Such atypical presentations may be associated with the altered proteolytic processing of other substrates rather than of APP. Both atypical and typical symptoms do not correlate well with Aβ plaques but rather NFT pathology [117]. In the past, individuals with atypical phenotypes were often diagnosed later but developments in biomarker testing is making earlier diagnosis possible [118].

Executive dysfunction is a common characteristic of LOAD, and may affect up to two-thirds of people with this diagnosis [119]; however, in a minority of case (possibly around a quarter) atypical presentations occur [19,26]. As mentioned, non-amnestic phenotypes also present in people with DS and they already perform more poorly pre-morbidly on tasks of executive functioning than other people with intellectual disabilities [120]; further among those with DSAD, performance is markedly poorer [121].

Personality changes, observed by carers [24], and often referred to as ‘behavioural and psychological symptoms of dementia (BPSD)’ [122] may be one of the first signs of the pre-clinical stages of DSAD [123]. BPSD comprises reduced empathy, emotional lability, apathy and social withdrawal [49,124], but not necessarily depression. The constellation of symptoms was first described in a longitudinal study of 108 individuals with DS, 68 of whom later developed dementia [125]. Later studies have supported these findings for example, personality changes were reported during the first stages of AD in a sample of participants (*n* = 331) from the DIAN cohort [24,126]. Changes in personality may, indeed, frequently occur before a decline in memory during the clinical stage of DSAD [124]. Despite the appearance of BPSD, which affects daily functioning [86,127] and is therefore easier to identify compared to memory changes, depression remains the major mental health issue affecting adults with DS [128]. As with other groups, the symptoms of depression may overlap with those of dementia [129].

The frequent presence of personality changes together with the executive dysfunction in DS [121] have suggested an early impact of dementia on the frontal lobes and/or their underlying neuronal circuits [130,131]. The pre-frontal cortex is an area that is underdeveloped and may be more susceptible to neurodegeneration in people with intellectual disability because of a reduction in cognitive reserve [132]. The findings of a neuroimaging study of fractional anisotropy, involving 20 adults with DS, half of whom had a diagnosis of DSAD, indicated a decrease in frontal white matter integrity that was associated with a decline in cognition [133]. A different perspective is that personality changes, at least, reflect the impact of negative life events, which are known to increase the risk of the development of AD [134]. Some of the available evidence does not support this view. For example, behavioural changes, such as disinhibition and problems in executive functioning (e.g., in planning, sequencing) are seen in the early stages of DSAD, while apathy appears more frequently only in the advanced stages [135]. In LOAD, in contrast, disinhibition is not seen until dementia advanced [136]. Further studies that take into account the relative underdevelopment of the frontal lobes in individuals are needed. Similarly, it will be important to consider the heterogeneity of the DS population: in people with severe or profound intellectual disabilities, early behavioural or personality changes are much more readily identifiable than changes in memory [124].

Personality changes that are so marked that they present as psychosis have also been reported among all groups with dementia. The symptoms affect about half of those with LOAD [137] but their prevalence remains uncertain in DSAD and the hereditary forms of dementia.

### 3.6. Neurological Symptoms

Neurological symptoms can present as the non-amnestic phenotype of AD. These tend to be more prominent and occur earlier in DSAD than in LOAD as shown by a 14 years longitudinal study of 77 persons with DS where it was found that neurological symptoms were more prevalent in DS with dementia than in non-demented DS (*p* = 0.0075) [15]. Similarly, presenting neurological symptoms are more common in ADAD [19] than in LOAD, especially in kindreds with a very early onset age of dementia (<40 years) [20].

The prevalence of neurological symptoms in ADAD was further corroborated by Voglein et al., 2019 [138]: motor symptoms had a prevalence of about 30% in ADAD, with increasing severity as the disease progresses [138]. In ADAD, there was a higher frequency of motor findings than in LOAD (28.4% versus 12.8%; *p* < 0.001) and the severity was also greater (mean UPDRS-III scores 2.0 versus 0.4; *p* < 0.001). Bradykinesia was particularly common in ADAD [138]. In the same study it was also found that the frequency and severity of the motor symptoms correlated positively with Aβ deposition within the basal ganglia. Indeed, there was a greater amount of Aβ in ADAD compared to LOAD (Pittsburgh compound B-standardized uptake value ratio 2.472 versus 1.928; *p* = 0.002). The heterogeneity in ADAD is noted as the frequency and severity of the motor symptoms were higher in PSEN1 mutations after codon 200 compared to those before codon 200 [138].

Some APP mutations in the Aβ coding domain (positions 692–694) have a clinical presentation at onset with epilepsy or resembling Dementia with Lewy Bodies [26]. Between 16% to 25% of PSEN1 mutation carriers, mainly with mutations in exon 8, after codon 200, show a non-amnestic phenotype [22,26] and present with behavioural and psychiatric symptoms, aphasia, visual agnosia, neurological signs and dysexecutive syndrome [26]. A retrospective study of 85 PSEN1 and 36 APP mutation carriers revealed the presence of spastic paraparesis, extrapyramidal signs, cerebellar ataxia and myoclonus [22]; but this is a rare presentation. Myoclonus was the most common symptom and with an increased the likelihood of developing seizures (*p* = 0.001 for PSEN1 and *p* = 0.036 for APP). Spastic paraparesis in some PSEN1 mutation carriers have been reported in association with cotton wool plaques [139,140], Lewy Body pathology (LBP) [26] or white matter and connectivity defects [141].

Myoclonus and seizures have both been associated with an earlier age of onset and more severe course of the disease in ADAD as reported in a systematic review [142]. A study of 5 families with APP duplications found that seizures were common (reported in 12 out of 21 participants) for this genotype [143]. In DS, epilepsy occurs at a frequency of up to 13% [144] and increases in those with AD diagnosis and up to 75% of adults with DSAD having epilepsy [145,146]. In contrast, the prevalence of epilepsy is lower in both LOAD (1.5–12.7%) and ADAD (2.8–41.7%) [147]. In a retrospective cohort study comparing 6430 individuals with DS to 19,176 controls, it was found that the incidence of epilepsy is elevated at all ages (*p* < 0.001), at least in Western countries [148]. Seizures occur as a bimodal distribution: early-onset epilepsy is associated with an absence of dementia, while myoclonic epilepsy in the late fourth decade of life or later is associated with the onset of AD [149]. In addition, seizures that had presented after the onset of dementia were associated with an accelerated functional decline. Such findings are supported by a study of 11 patients with DS or AD with myoclonic epilepsy [150]. Nevertheless, a review of the symptoms of people with DS in Japan and China, where epilepsy is less prevalent among all the population, indicates [150], high rates of epilepsy are not found always found. The difference could potentially be explained by an environmental impact.

The cause of seizures remains uncertain. No association has been found with APOE ε4 status in ADAD [22], but in mouse models, both TAU [151] and Aβ deposition [152] have been related to epileptiform activity. Furthermore, as reviewed in [153], DS neurodevelopmental abnormalities, including structural and biochemical alterations, may play a role. It is also possible that the increased prevalence is actually due to an over-reporting of the atypical symptoms for people with ADAD and DSAD or that their younger age at onset allows them to survive until the more severe stages of the disease than in LOAD [20]. Another explanation is that this is related to increased plaque load due to Aβ42 deposition which would explain common observation of seizures in DS, dupAPP and certain other APP mutations [50]. Alternatively, seizures may be associated with a GABA/Glutamate (inhibition/excitation) imbalance. In fact, GABAergic dysfunction has been seen in AD [154] and is altered in DS [155].

Another characteristic of DS are the musculoskeletal disorders which appear earlier than in the general population, probably due to the accelerated ageing, and which can trigger abnormal posture and unsteady gait. This decline has also been associated with LOAD in its pre-clinical stage [156].

## 4. Neuroimaging and Neuropathology

A cross-sectional functional MRI connectivity study of 83 mutation carriers from the DIAN cohort, harbouring either APP, PSEN1 or PSEN2 mutations, reported that the clinical diagnosis of AD can be preceded by many years by functional disruptions of the default mode network (DMN) in ADAD [157]. Similar changes in resting state network, though occurring later, are found in LOAD [157,158]. Altered default mode connectivity was also observed in adults with DS with detectable fibrillar Aβ as measured by positron emission tomography (PET) using PiB as the tracer [159]. There are different explanations for the functional changes described. In ADAD, pre-clinical white matter degeneration may contribute to DMN disruption [160]. A similar explanation may be applicable to DS [12]. In DSAD, the preclinical stage could be characterised by different compensatory responses [161]. For example, an increase in glucose uptake was described in a cross-sectional PET study that compared the regional cerebral glucose metabolic rates (GMR) of 17 DS with both demented (*n* = 10) and non-demented controls (*n* = 24) during a cognitive task [162].

In AD, a classification or staging of the disease using biomarkers has been developed: the A/T (N) system which describes the presence or absence of Aβ (A), tau (T) and neurodegeneration (N) without taking into account the order of onset of biomarker change. [163,164]. In order to unify criteria among research centres, the application of the A/T/N system has been examined. Using this description of the pathological process, it has been established that the biochemical changes in DSAD, ADAD and LOAD are similar both in direction and magnitude. In addition, the three types of dementia have similar atrophy, hypometabolic and Aβ deposition maps. Figure 3 (taken from Fortea et al. 2017 [165]) shows changes in several metrics in DSAD. The pattern of changes in neuroimaging and biomarkers, which is similar in LOAD and ADHD [166,167] indicates the following order: an increase in amyloid deposition, followed by a decrease in glucose metabolism, and finally atrophy, across many cortical regions. In all three conditions, this similarity in the order of change of biomarkers is referred to as the “Jack curve” [168]. The findings indicate that the A/T/N system has potential as a screening tool in AD.

The Thal Aβ phases (or staging), show the progression of Aβ deposition in the brains of AD patients at different points in the disease. In AD, the deposition of Aβ increases as a function of estimated years from expected symptom onset, with a characteristic cortical deposition in LOAD that is shared by ADAD, though it does not follow the exact same pattern [167]. A PIB-PET study of 128 ADAD mutation carriers found an early deposition in the precuneus [169] and in the posterior cingulate gyrus [170,171], that is greater in those with PSEN1 mutations after codon 200, as well as for those carrying an ε4 allele [20]. Similar brain amyloid deposition patterns are seen in DSAD, whereby there are neuritic plaques in the parieto-temporal, precuneus, posterior cingulate, and frontal regions similar to LOAD [172]. A (18F) FDDNP PET amyloid study showed that plaque density was higher in the brains of 19 adults with DS without dementia when compared with 10 patients with LOAD, and its deposition was greater in the frontal and parietal cortex [173].

In LOAD, there is no initial striatal deposition as measured by PET amyloid-ligands. Diverging from LOAD, 11C-PIB-PET imaging studies of 49 adults with DS [174], and of 346 ADAD individuals [167] have report that, in genetic AD there is an increased Aβ deposition in subcortical regions, especially in the striatum. The significance of this observation, which is universal in ADAD and in DSAD, however, is unclear. It could potentially be a consequence of reduced clearance mechanisms in the striatum in genetic AD. Figure 4 shows a map of the progression of Aβ binding to different areas of the brain. As in the E280A PSEN1 mutation there seemed to be no binding to the striatum [171], which highlights the heterogeneity of ADAD. The degeneration pattern is not equally followed in all brain areas in ADAD, as some regions show different susceptibility in a mutation-dependent way [166,167]. Whilst there is no initial striatal Aβ deposition in LOAD, as the disease progresses, there will eventually be deposition of Aβ there [175,176].

The increased Aβ is associated with the allele ε4 is seen in the general population [177], but its role in ADAD [142] and DS was controversial when comparing the results obtained by the previously cited studies [178,179,180]. However, Bejanin et al. [36] demonstrated in a DS cohort of 464 that the ε4 allele does indeed influence the earlier onset of the presence of amyloid biomarkers.

Aβ42 plaques are more prevalent than Aβ40 plaques at all ages in ADAD and DS [12,181]. In DS, concentrations of Aβ40 and Aβ42 are higher compared to LOAD [182]. However, the Aβ42/Aβ40 ratio shows no difference between the two conditions, demonstrating similar processing and deposition of Aβ40 and Aβ42 [182] during the “transition” phase to greater plaque numbers. Novel mass spectrometry techniques that accurately detect Aβ in LOAD have not yet been reported in DS [183]. In ADAD, a study comparing 18 children with the PSEN1 E280A mutations with 19 noncarriers showed that ADAD mutation carriers had increased plasma Aβ42 levels when compared to controls (*p* < 0.001) [171]. Exceptions, however, are notable. For instance, in the APP mutations—Dutch mutation E693Q or the Italian mutation E693K—the Aβ42/40 ratio is reduced [50]. These exceptions may, at first sight, seem to contradict the amyloid hypothesis. However, a reduction in the ratio does not necessarily mean a reduction in the absolute level of Aβ. A reduction in the ratio can be due to increases in Aβ40 and/or reduction in Aβ42, but with an overall increase in total Aβ.

Different studies have reported a range of associations between changes in Aβ levels and dementia status. A prospective study of 530 individuals found a positive association between AD and alterations in the concentration of Aβ42 but not Aβ40 [184], a 10-year follow-up case–cohort study of 6713 participants found that both concentrations were affected [185] and a longitudinal study of 237 ADAD [186] reported no changes in Aβ concentration. These different results are also seen in DSAD [12]. A prospective study of 204 adults with DS found alterations just in Aβ42 concentration related to AD [179]. Another project with 225 adults with DS found an increased Aβ42 and decreased Aβ40 [187]. Similarly, lower levels of Aβ40 were found in 44 participants with DS and dementia compared with 83 individuals with DS and no dementia [188]. A study of 506 DS found increases in both peptides [189]. In a meta-analysis Alhajraf et al. 2019 [190] showed that Aβ40 levels in the plasma increased and that plasma Aβ40 levels could potentially act as a marker for predicting AD in DS. Some studies reported no changes in Aβ concentration as was the case reported in a longitudinal study with 78 DS participants [178], and another study with 60 [180]. It is critical to point out, however, that is no agreement about the best kind of plasma biomarker assay. The contradictory findings may reflect the use of different assays in different studies.

CSF soluble Aβ42 levels in ADAD [186] and DS [191] are high early in life and then decline rapidly as they presumably start depositing into plaques [192]. ADAD mutations, an ApoE ε4 allele and the chromosome 21 trisomy have been for a long time associated with a higher density of plaques, as described by in a report of the distribution of senile plaques in AD [193]. In a post-mortem study of 60 patients with ADAD, neuritic plaques were found at a higher level compared to 120 participants with LOAD [77]. In PSEN1 mutations, particularly those after codon 200, cotton-wool plaques seem to be more frequent [194]. In DS, a histological study described that, not only the density of plaques is altered, but also its form: for 12 DS cases, amyloid plaques were larger and had a more amorphous morphology than those present in 10 LOAD cases [195]. However, certain studies have shown that Aβ plaques and neurofibrillary tangles in DSAD are similar in appearance to LOAD [196].

Following Aβ deposition metabolic deficits start; in a retrospective study of 146 with DS metabolism was associated with a tendency from aerobic (lower lactic acid levels) towards more glycolysis and subsequent lactic acid fermentation metabolism [197]. Hypometabolism drives widespread cell stress that leads to neuronal loss, especially in the precuneus of ADAD mutation carriers, as described by a FDG-PET study of 20 ADAD versus 20 LOAD subjects and in the posterior cingulated cortices of ADAD [198] and DS [162]. The hypometabolism pattern in DS is similar to that in LOAD involving the parietal, precuneus and posterior cingulate [199,200]. Some studies have reported hypermetabolism in DS, especially in young adults as a consequence of less efficient glycolysis or potential compensatory mechanisms [162,201,202].

Atrophy tends to occur in AD. Serial MRI scans of 66 ADAD participants compared to 28 controls [102] showed that approaching the age of onset of dementia, there is increased ventricular volume and hippocampal atrophy. In comparison with LOAD, a prospective MRI study of 12 patients showed that atrophy has an accelerated course in ADAD [203]. Atrophy was also seen in the Down Syndrome Biomarker Initiative, in which 12 DS adults took part for the volumetric study [204]. The atrophy pattern in DS is similar to that in LOAD, involving posterior dominant cortical thinning with atrophy of hippocampus, thalamus, and striatum [205,206]. At this stage, the brain reserve is not sufficient to compensate for the deficits, and the symptoms that characterise and are associated with a clinical diagnosis of dementia become evident. In DS, atrophy seems to be less intense perhaps suggesting that chromosome 21 may encode a gene that is neuroprotective when triplicated [40]. Another possibility, however, is that the more limited atrophy may simply reflect smaller whole brain volume in DS, as described in a study from 1991 of 7 adults with DS [207].

Moreover, the regional distribution of tau (both from PET studies and histopathology) is broadly similar in DS and LOAD [208]. The pattern accords with the tau Braak staging system [209,210]. Braak staging shows the distribution of tau within the brain at different points in AD. Amyloid deposition precedes tau pathology in the cerebral cortex and subcortical nuclei of the forebrain, akin to LOAD [208]. In addition, and as in LOAD, tau pathology is first seen in the entorhinal cortex of the temporal lobe [208]. Importantly, the early involvement of brainstem monoamine producing neuron systems has also been described in DS [208]. The distribution of tau in ADAD is also similar to that in LOAD with the areas of high tau PET binding overlapping with those in LOAD [211]. Similar to LOAD, tau is present in the temporal and parietal regions [211]. However, despite similar cognitive impairment in the two conditions, there seems to be greater cortical involvement and higher levels of binding in ADAD [211].

Later in life, there is an increase in CSF tau concentration levels in LOAD [212] and DS [192], while in ADAD, there is a decrease in its levels, according to a longitudinal study of 411 individuals [213]. In fact, CSF and plasma biomarkers, notably neurofilament light and tau181, have been shown to have good potential for predicting AD in DS [165,214,215]. As plaques continue to develop, tangles form, supporting the hypothesis of Aβ as the driving force in AD [216]. The time lag between Aβ pathology and NFT pathology is similar in ADAD and DS [217]. The distribution of plaques and NFT is similar between the three groups [40] and in a 15 year prospective study of 92 hospitalised adults with DS, it was shown that NFTs also correlate better with dementia than with amyloid deposition [218].

The different subtypes of LOAD and the neuroimaging findings and biomarker changes for these subtypes reveal respective differences [219]. For instance, four distinct trajectories of tau deposition have been demonstrated in LOAD [220], and in terms of the temporal complexity, three subtypes can be classified [221]. In discussing the neuroimaging and neuropathology of LOAD, the focus throughout this review has been on the LOAD that presents with amnestic symptoms.

## 5. Co-Pathologies

AD is often associated with co-pathologies such as Lewy body pathology (LBP) and TDP-43 pathology [219]. The number of co-pathologies increases with age [222]. These can affect both diagnosis and disease progression. Although they are not the focus of this review, they are briefly considered here.

### 5.1. Lewy Body Pathology

Lewy bodies (LB) are pathological aggregates of proteins in the brain. In an autopsy study, it was found that the number of LB deposits in DS increases with age [223]. Clinically, this can manifest as LB dementia but it is rare in DS, but the first case was reported in 2010 [224]. In both LOAD and ADAD, LB dementia is a frequent comorbidity as has been reported in both the ADNI and DIAN cohorts [225]. In ADAD, LB dementia is more common than might be expected to occur in a young population; we speculate this may also be true in DS. The explanation may lie in APP mismetabolism [77], following the hypothesis that LB could be induced by the accumulation of Aβ [20]. LB are less extensive in ADAD compared to LOAD [77], with the prevalence in LOAD being between 6 and 39% [226].

### 5.2. TDP-43 Pathology

TDP-43 pathology co-occurs in all three forms of AD and seems to be associated with the development of amnestic phenotypes [227]. The distribution of TDP-43 affects the amygdala and hippocampus more than the neocortical regions, where there is an absence of TDP-43 [228]. However, the frequency at which TDP-43 pathology cooccurance in the three forms differs. TDP-43 pathology seems to be more prevalent in LOAD compared to ADAD and DS [227]. In LOAD, limbic predominant age-related TDP-43 encephalopathy neuropathological change (LATE-NC) occurred in 57% of cases and correlated with faster disease progression and cognitive impairment [229]. This, and similar, findings have led to the hypothesis that TDP-43 pathology may be a side effect of ageing rather than of AD [227].

## 6. Other Similarities and Differences

There are other similarities and differences observed in neuropathology between the LOAD and in DS that have not yet been thoroughly examined. These include impairments of the noradrenergic [230] and immune systems [231], NGF metabolism [232,233], enhanced inflammation, and increased oxidative stress [231]. Such features warrant further investigation since although APP plays an important role, there are many oxidative [234,235] and inflammatory [236] genes on chromosome 21 that overexpressed in DS. These genes may cause a neuroinflammatory states that as in LOAD, may result in a self-amplifying cycle that leads to the development and/or maintenance of AD [237]. In common with LOAD [238], DS shows upregulation of inflammatory response, as seen by elevated levels of cytokines [239] or the association of proteins of the complement cascade to Aβ plaques [240]. Moreover, in DS, there is also increased microglial activation with increased age compared with healthy controls [241]. Nevertheless, there appear to be differences: for example, reflecting the distinct profile of microglial states in LOAD, the neuroinflammatory phenotypes of the two conditions are not the same [242].

There are other areas to investigate. First, there is neopterin, a marker for cell-mediated immune activation and inflammation. Higher plasma concentrations have been found in individuals with DSAD compared with those with DS without dementia [243]. At the same time, neopterin levels in urine predict cognitive decline in people with DS over time [244]. A different focus is IL1β. Levels of IL1β levels were higher in DS compared to LOAD [182]. Moreover, IL1β was correlated with t-tau, suggesting that it may be associated with neurodegeneration [182]. It should be noted, however, that high levels of IL1β in DS may simply reflect increased prevalence of autoimmune conditions and/or heightened vulnerability to infections in people with the condition [182], presenting as autoimmune diseases [245] such as hypothyroidism [246]. Secondly, the development of DSAD may also be facilitated by increased vulnerability to certain infections. For example, the severe immunodeficiency in the salivary IgA for DS [247] along with the increased susceptibility for impairments of DS’s gingival fibroblasts [248] may be related to the higher prevalence of severe early-onset periodontal diseases in DS [249]. Periodontitis caused by Porphyromonas gingivalis has been linked to an increased risk of developing AD [250,251].

In DS, mitochondrial dysfunction and its consequent higher levels of reactive oxygen species (ROS) have been reported [12]. Oxidative stress is enhanced in DS compared to the general population and may contribute to increased lipid and protein peroxidation that promotes increase in DSAD [252]. Moreover, the level of superoxide dismutase enzymes, which are antioxidant enzymes, could predict memory decline over time in DS [253]. Further corroborating the relationship between AD and oxidative stress is the suggestion that changes in iPF2α, a marker of oxidative stress, are correlated with cognitive decline [254]. Increased oxidation has also been linked with increased Aβ production [255,256]. Moreover, as in LOAD, the mitochondrial DNA (mtDNA) mutation rate is particularly high in the brains of people with DSAD [257]. All these processes could contribute to accelerating the onset of dementia in DS [235]. It has been suggested however that other researchers suggest that the systemic accumulation of Aβ or the lipid peroxidation may alter the mitochondria integrity with the resulting dysfunction leading to a self-amplifying loop [234].

## 7. Conclusions

In the genetic cases of AD (DSAD and ADAD), there are life-long neuropathological changes that are not present in the general population until the early stages of AD. In the majority of cases in all the three groups compared, the clinical appearance of dementia starts with memory deficits. However, in ADAD and DS, there are more non-amnestic phenotypes. Moreover, in those with DS and most forms of ADAD, there is also more severe CAA and increased neurological symptoms compared to LOAD. Both people with DS and carriers of ADAD mutations show a higher and earlier brain Aβ load, as well as an increased accumulation of Aβ plaques and NFTs. The increased Aβ deposition probably accounts for the early age of onset described in these hereditary AD cases. Though the magnitude and direction of changes in the three conditions are generally similar, there are some differences. In contrast to LOAD, ADAD and DSAD have an increased initial accumulation in the subcortical regions, particularly in the striatum. Table 1 recapitulates the similarities and differences between the three forms of AD, highlighting the heterogeneities, particularly in ADAD and age associated comorbidities There remain gaps in our understanding the reasons for the differences in clinical presentation and the genotype-phenotype relations of these conditions. With on-going large longitudinal clinical studies of DS, ADAD and LOAD (all funded by the National Institutes of Health), namely ABC-DS (Alzheimer’s Biomarkers Consortium-DS), DIAN and ADNI, respectively, data from these studies are beginning to provide a fine-grained characterisation and understanding of AD. Concurrently, biochemical and cellular understanding is being made possible by studying known mutations of the genetic forms of the disease aided by the use of induced pluripotent stem cells, organoids and gene-editing techniques. Importantly, a multi-scale understanding (for example, molecular changes and their impact on tissue pathology or cognition) and the generation of new hypotheses is likely when individual case studies are comprehensively investigated.

There were multiple difficulties encountered throughout this review. The main limitation was the lack of head-to-head comparisons. It is also important to highlight that many features of DSAD, especially those defined decades ago have not been replicated. It is difficult to extrapolate which features belong to ADAD in general, rather than to specific mutations. For the mutations in a particular gene or from the same kindred, we cannot discard that the characteristics studied belong to the specific mutation of those patients, as would happen with a case-report, rather than being a shared feature of mutations in that gene. This is especially the case in PSEN2, which is rare and so cohorts studied tend to be small. Taking all these considerations into account, we consider that much is left to be explored and more research is a must. Understanding the pathology behind the different forms of AD and the differences and similarities with LOAD will hopefully allow us a deeper insight into the causes of AD and potentially lead to the development of new targeted and personalised therapies.

## Figures and Tables

**Figure 1 jcm-10-04582-f001:**
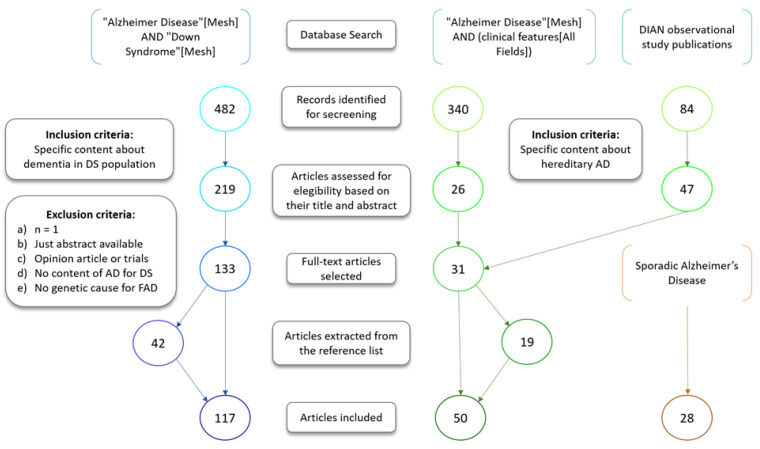
Search methodology undertaken for the review. AD, Alzheimer’s Disease; DS, Down syndrome-associated; FAD (Familial Alzheimer’s Disease), DIAN, the Dominantly Inherited Alzheimer Network.

**Figure 2 jcm-10-04582-f002:**
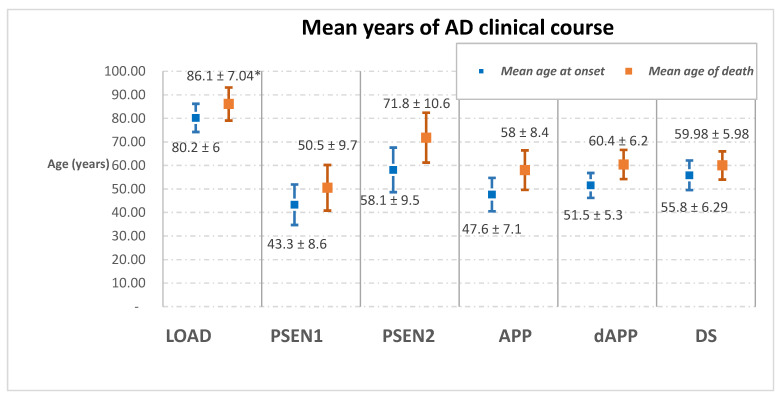
Schematic representation of the average duration of Alzheimer’s clinical course that has been reported for the sporadic form of the disease (LOAD) [67], the familial one (PSEN1, PSEN2, APP and dAPP) [68]; and AD in the DS population (DS) [61]. * The mean age of death for LOAD was calculated from the mean age at onset plus the average survival; LOAD = Late-onset Alzheimer’s disease.

**Figure 3 jcm-10-04582-f003:**
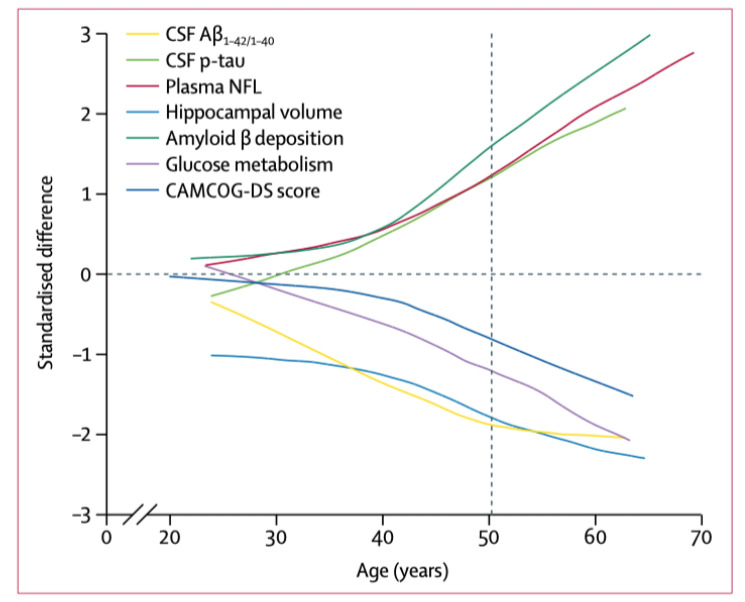
Integrated model of the natural history of AD in individuals with DS. Changes in structural, metabolic, and biochemical biomarker parameters as a function of age by use of the standardised differences between participants with DS and cognitively healthy controls fitted with a locally estimated scatterplot smoothing curve. Positive standardised differences represent higher biomarker values in participants with DS than in euploid controls, and negative values represent lower biomarker values. The vertical dashed line at 50·2 years represents the age at the expected symptom onset (median age of prodromal Alzheimer’s disease diagnosis). CAMCOG-DS (Cambridge Cognitive Examination for Older Adults with Down Syndrome) comprises brief neuropsychological tests used to measure cognitive decline. NFL = neurofilament light chain. p-tau = tau phosphorylated at threonine 181. From Fortea et al. [165] 2020. CSF (cerebrospinal fluid).

**Figure 4 jcm-10-04582-f004:**
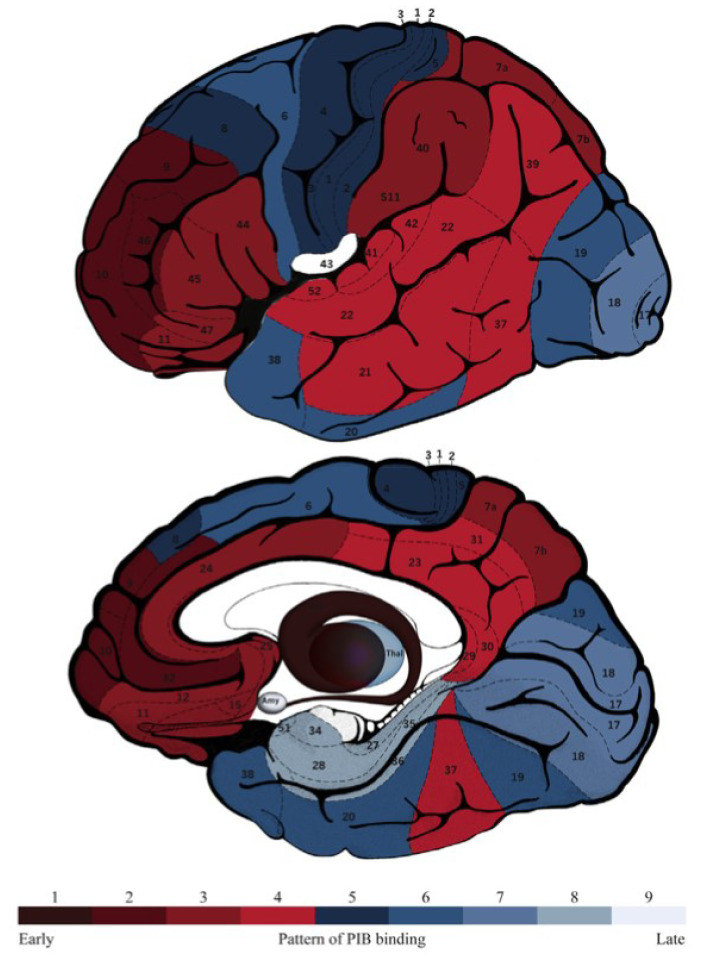
A schematic brain map of numbered Brodmann areas and subcortical regions of interest colored according to the PIB staging model in DS, where shade 1 denotes the area affected first (i.e., the striatum) and shade 9 the area affected latest (the amygdala). Abbreviations: thal, thalamus; amyamyg- dala; PIB, Pittsburgh compound–B. From, Annus et al., 2016 [174].

**Table 1 jcm-10-04582-t001:** Summary of the comparison between the three forms of AD.

	LOAD	ADAD				DSAD
Clinical Features						
Mean age of onset	80.2 ± 6	PSEN1	PSEN2	APP	dAPP	55.8 ± 6.29
		43.3 ± 8.6	58.1 ± 9.5	47.6 ± 7.1	51.5 ± 5.3	
Mean age of death	86.1 ± 7.04	50.5 ± 9.7	71.8 ± 10.6	58 ± 8.4	60.4 ± 6.2	59.98 ± 5.98
Cerebral Amyloid Angiopathy	Low	High (with higher prevalence in dAPP)				High
Amnestic phenotypes	Early episodic memory loss Early loss of visuoperceptual skills Attention deficits Language deterioration	Early episodic memory loss Late loss of visuoperceptual skills Attention deficits Language deterioration (depending on the particular mutation)				Early episodic memory loss Early loss of visuoperceptual skills Attention deficits Language deterioration Decline in functional skills
Non amnestic phenotypes (e.g: behavioural changes and executive dysfunction)	Less Common	More Common				More Common
Neurological symptoms (e.g: epilepsy, myoclonus, spastic paraparesis, cerebellar signs)	Less Common	More Common				More Common
Neuroimaging and Neuropathology						
Altered default mode connectivity	Present	Present				Present
Biochemical changes	Similar in magnitude and direction	Similar in magnitude and direction				Similar in magnitude and direction
Aβ deposition map	Similar	Similar				Similar
Initial Striatal Aβ	Absent	Present				Present
Hypometabolism map	Similar	Similar				Similar
Atrophy map	Similar	Similar (but accelerated)				Similar
Distribution of tau	Similar	Similar				Similar
CO-PATHOLOGIES						
Lewy body pathology	Common	Most common				Rare
TDP-43 Pathology	More common Similar distribution	Less Common Similar distribution				Less Common Similar distribution

AD, Alzheimer’s disease; LOAD, Late-onset Alzheimer’s disease; ADAD, Autosomal Dominant Alzheimer’s disease; DSAD, Down syndrome-associated Alzheimer’s disease.

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
