# Peer review of "The Clinical and Neuropathological Features of Sporadic (Late-Onset) and Genetic Forms of Alzheimer’s Disease"

_jcm, 2021, doi:10.3390/jcm10194582_

Round 1

Reviewer 1 Report

This systematic review deals with a much-needed review of our understanding of the genetics of Alzheimer’s disease (AD) in 3 major forms of AD: the most common sporadic AD and 2 rare forms of AD in high risk individuals (i.e., autosomal dominant AD and Down syndrome). This review further examines the complexities involving “genotype-phenotype relations” in AD, where genotypic heterogeneity along with phenotypic heterogeneity compound the difficulties in understanding the relationship.

Some issues. 

  • This paper provides a much needed and timely overview of the differences and similarities between the special populations with AD. While the review is comprehensive, touching upon multiple aspects of AD, it would have been better if there was a separate discussion of the genetic factors by each of the 3 groups, which may impact the clinical/neuropathological features. This probably reflects the current state of knowledge, where there haven’t been extensive genomic studies conducted in 2 other high groups as was done in the general sporadic form of AD.
  • It would be desirable to highlight the current gaps in the literature and suggest future research, which may enhance our understanding.
  • Table 1 nicely summarizes the similarities and differences among the 3 groups. It would benefit the readers if a more precise language was used (or explained in the footnote), rather than using terms like “More/Less Common,” etc, or listing neurological symptoms for clarification, for example.
  • Line 78. A further explanation is needed when the authors states that “survival bias influences these ”
  • It would also be important to note that over 300 different PSEN1 mutations are present and associated phenotypic variation is substantial.
  • It is well known that many genetic mutations have been reported for 3 known autosomal dominant genes. To date, 70+ different mutations in APP have been reported, 300+ mutations in PSEN1, and 60+ mutations in PSEN2. Phenotypic variability associated with each of these variants is likely to be tremendous. Perhaps it is worth mentioning somewhere potential difficulties these heterogeneities impact genotype-phenotype relations when significant efforts are being made to push toward precision medicine.

Minor issues.

  • In Table 1, the mean age of death (55.8) is earlier than the mean age at onset (59.98) in DSAD. Most likely, the numbers are switched or it is a typographical error.
  • Typo in page 2, line 59-60: “…It is caused by an extra copy of chromosome 21 in c.95% of cases;…”
  • Line 192. It would be helpful to indicate how many additional papers were included from examining the references of other papers.
  • Line 521. References are missing.

Author Response

REVIEWER 1

  • We thank the reviewer for agreeing that this is a much needed and timely overview. The reviewer however, requested “a separate discussion of the genetic factors by each of the 3 groups”. We did not write separate sections to emphasise the need to discuss the 3 groups “side-by-side” as we believe that this approach was more effective when genetic causes are contrasted with the sporadic or late onset causes. We also wanted to keep Down syndrome as a genetic cause as “centre-stage” as it is often overlooked or under emphasised possibly because of the added intellectual disability and stigma attached to this population. Also, as pointed out by the reviewer, there are less studies of the genetic nature that have been carried out in the two genetic groups as in the late onset group. Although we therefore kept the same order, we took the opportunity to clarify the writing in places where it was unclear.
  • We have added sentences to highlight the current gaps in the literature and included suggest future research which may enhance our understanding; this was an excellent suggestion by the reviewer. Thus under conclusion we have now added:

There remain gaps in our understanding for the reasons for the differences in clinical presentation and the genotype-phenotype relations of these conditions. With on-going large longitudinal clinical studies of DS, ADAD and LOAD (which are all funded by the National Institutes of Health), namely ABC-DS (Alzheimer’s Biomarkers Consortium-DS), DIAN and ADNI respectively, these are beginning to provide a fine-grained characterisation and understanding of AD. Whilst the biochemical and cellular understanding is being made possible by studying known mutations of the genetic forms of the disease aided by the use of induced pluripotent stem cells, organoids and gene-editing techniques. Importantly, a multi-scale understanding (for example, molecular changes and their impact on tissue pathology or cognition) and the generation of new hypotheses is likely when individual case studies are comprehensively investigated.”

  • For Table 1 the reviewer wanted “more precise language” instead of “more/less common”. The reason we chose to use imprecise phrasing was that the conditions being relatively rare reduces the number that can be studied negatively impacting on the generalisability or statistical confidence of quantitative analyses. In the table, however we have added a more examples of neurological features commoner in the genetic forms (myoclonus, paraparesis and cerebellar signs).
  • Line 78. We have clarified “survival bias influences these ” by adding a new sentence (now on line 81 and 82).
  • As, “It would also be important to note that over 300 different PSEN1 mutations are present and associated phenotypic variation is substantial”; we have now emphasised this in the revised text as we agree this is an important point (see lines 80 to 92). The variety is also an opportunity to study genotype-phenotype relations; we mention this in the revised conclusion.
  • The reviewer highlights the need to emphasis the large number of APP and PSEN mutations and also, “perhaps it is worth mentioning somewhere potential difficulties these heterogeneities impact genotype-phenotype relations when significant efforts are being made to push toward precision medicine.” We have now highlighted the variety and number more in the revised text and commented on the impact on precision medicine (lines 80 to 92).
  • MINOR ISSUES:
  • In Table 1, the mean age of death was corrected.
  • Typo in page 2, line 59-60:we have corrected this.
  • Line 192. Additional paper number now added to line 209.
  • Line 521. References are missing-reference now added.

Reviewer 2 Report

The authors aim to compare the clinical and neuropathological manifestations of Alzheimer's disease in three groups of patients: (1) sporadic individuals, (2) inherited form with autosomal dominant trait and (3) AD associated with Down syndrome. The manuscript is written in an orderly way. Neuroimaging and neuropathological chapter is described in detail, which is consistent with the title. Furthermore, a table showing the differences and similarities between the different forms of AD is a benefit and increases the readability of the paper. 

Author Response

We are very thankful for the reviewer for appraising and approving our article. We believe that it will be an important contribution to the literature. 

We have made some changes as suggested by the other reviewers (uploaded).

Reviewer 3 Report

Some issues:

1) Abstract: Does not include the purpose for conducting this review. Also, it should state the key findings and conclusions. 

2) The introduction should be restructured. Section 1.1 should be removed or merged within the introduction.  Perhaps it is not needed, given the focus of this review to compare different forms of AD. Introduction should end with the purpose and aim of the review, from where the methods section should start.

4) The first paragraph of methods section is poorly explained. A read through makes one confused. Should be explained clearly, and moreover PRISMA flow charts are missing.

5) According to authors DS the best population to conduct AD studies. This is a wrong statement. Please specify which AD studies, because AD studies will include even those focused on EOAD without DS.

6) Grammatical error: line 67

Author Response

We are grateful to you for this opportunity to improve the paper. We respond to the individual points you made below.

1) We have improved the abstract as suggested. 

2) We have improved and restructured the introduction and merged section 1.1 as suggested, but we felt that removing parts of it would limit the comprehensiveness of the review that we had intended.

4) We have improved the methods section and added a PRISMA flow chart as requested.

5) “According to authors DS the best population to conduct AD studies. This is a wrong statement. Please specify which AD studies, because AD studies will include even those focused on EOAD without DS.” We have removed this statement and instead stated made a more nuanced statement in line 128 of the revised version:

Clinical studies in DS have some advantages as there is both a high risk and greater predictability of developing DSAD, and the AD neuropathology seems more homogeneous than in LOAD.”

6) Grammatical error: line 67-this has been corrected.

Please note that we have also incorporated improvements suggested by the other reviewers (see revised paper).

Round 2

Reviewer 3 Report

The paper looks good now! Thanks for making the changes.